# Understanding the Nature of the Long-Range Memory Phenomenon in Socioeconomic Systems

**DOI:** 10.3390/e23091125

**Published:** 2021-08-29

**Authors:** Rytis Kazakevičius, Aleksejus Kononovicius, Bronislovas Kaulakys, Vygintas Gontis

**Affiliations:** Institute of Theoretical Physics and Astronomy, Vilnius University, Sauletekio al. 3, 10257 Vilnius, Lithuania; aleksejus.kononovicius@tfai.vu.lt (A.K.); Bronislovas.Kaulakys@tfai.vu.lt (B.K.)

**Keywords:** long-range memory, 1/*f* noise, absolute value estimator, anomalous diffusion, ARFIMA, first-passage times, fractional Lèvy stable motion, Higuchi’s method, mean squared displacement, multiplicative point process

## Abstract

In the face of the upcoming 30th anniversary of econophysics, we review our contributions and other related works on the modeling of the long-range memory phenomenon in physical, economic, and other social complex systems. Our group has shown that the long-range memory phenomenon can be reproduced using various Markov processes, such as point processes, stochastic differential equations, and agent-based models—reproduced well enough to match other statistical properties of the financial markets, such as return and trading activity distributions and first-passage time distributions. Research has lead us to question whether the observed long-range memory is a result of the actual long-range memory process or just a consequence of the non-linearity of Markov processes. As our most recent result, we discuss the long-range memory of the order flow data in the financial markets and other social systems from the perspective of the fractional Lèvy stable motion. We test widely used long-range memory estimators on discrete fractional Lèvy stable motion represented by the auto-regressive fractionally integrated moving average (ARFIMA) sample series. Our newly obtained results seem to indicate that new estimators of self-similarity and long-range memory for analyzing systems with non-Gaussian distributions have to be developed.

## 1. Introduction

Many empirical data sets and theoretical models have been investigated using the tool of spectral analysis. Many researchers across different fields find the power spectral density (abbr. PSD) of the $1/f^{\beta }$ form (with 0.5≲β≲1.5) to be of a particular interest [1,2,3,4,5,6,7,8,9,10], both because of its apparent omnipresence and the implication of slowly decaying autocorrelation, which indicates the presence of the long-range memory phenomenon. Long-range memory is also one of the established stylized facts of the financial markets [11,12,13,14,15,16,17,18,19]. Consequently, as our group was investigating 1/f noise [20,21,22,23], we have become naturally interested in the rapidly growing field of econophysics. The term “econophysics” was coined by H. E. Stanley in the Statphys conference in Kolkata in 1995 [24]. Over the last three decades, econophysics has matured both from the theoretical and the applied perspectives. Here, we review mostly our own and directly adjacent approaches, and we would like to recommend a couple of broader reviews, which can be found in [25,26].

Our first publications were devoted to the modeling of the financial markets [27,28]. In those works, we have considered trades occurring in the financial markets as point events driven by a point process proposed in [21,22,23]. Thanks to the organizers of the international conference Applications of Physics in Financial Analysis 4, held in Warsaw in 2003, we were able to present our findings to econophysicists. Our first results, inspired by interaction with the participants of the APFA 4 conference, have been published in [29,30]. We presented our ideas in a more general context of complex systems in [31,32].

Later, we took part in the COST Action P10 “Physics of Risk” and the follow-up COST Action MP0801 “Physics of Competition and Conflicts”. Bronislovas Kaulakys and Vygintas Gontis were executive committee members of both COST Actions, while the other group members gave talks and poster presentations during the annual meetings and helped organize an annual action meeting in Vilnius in 2006. This COST action meeting has helped us embrace econophysics and be recognized as econophysicists.

While it may be natural to see trades in the financial markets as point events [27,28,29,30], modeling volatility and return as a point process was not as straightforward. We have developed our approach further by abstracting the point process away and considering a continuous framework of Langevin stochastic differential equations (abbr. SDEs). First, we have shown that the continuous interpretation of the point process model works well for trading activity [33]; thus, we have refined the SDE approach with model for volatility and return [34,35,36,37,38]. Interestingly, similar SDEs can be derived from a simple agent-based model (abbr. ABM) [39,40], too. With time, we have developed more complicated ABMs to account for the separation of time scales and order flow [41,42]. We have even branched out into sociophysics [43,44,45,46] as we have understood that the herding ABM we used to model the financial market is essentially equivalent to the well-known voter model [47,48,49].

For 10 months (in 2015 and 2016), Vygintas Gontis, with the support of the Baltic American Freedom Foundation, has stayed as a visiting researcher at the Center of Polymer Studies of Boston University. Discussions with the founding fathers of econophysics, H. E. Stanley, professors Sh. Havlin, B. Podobnik, and S. Buldyrev, resulted in a paper [50]. Together, we have considered volatility return intervals (term inspired by the studies [51,52,53,54]) of the financial time series at various time scales. In the paper, we have shown that the time intervals between large financial fluctuations is distributed according to a power–law probability density function (abbr. PDF) $p\left(\tau \right)∼\tau^{-3/2}$ [50]. The same distribution arise in our models and from many other one-dimensional Markov processes [55], while the long-range memory process would exhibit a different distribution, such as $p\left(\tau \right)∼\tau^{2-H}$, which is a well-known result for the fractional Brownian motion (abbr. FBM) [56].

Here, we provide an overview of our approach to understanding and modeling the long-range memory phenomenon in financial markets and other complex systems and share our most recent result. In Section 2, we introduce the original point process and discuss how to derive a non-linear SDE, which can reproduce the long-range memory phenomenon. We also discuss numerous extensions of both the point process model and non-linear SDE. Next, in Section 3, we show how we can obtain a similar SDE from a simple herding ABM. Following the overview, we also present a novel result, which concerns understanding the nature of the self-similarity and long-range memory phenomenon from the perspective of fractional Lèvy stable motion (abbr. FLSM) and auto-regressive fractionally integrated moving average (abbr. ARFIMA) time series. In Section 4, we tested various long-range memory estimators such as mean squared displacement, method of absolute value estimator, Higuchi’s method, and burst and interburst duration analysis on fractional Lèvy stable motion (ARFIMA(0,d,0) time series). Finally, in Section 5, we share our future considerations.

## 2. The Multiplicative Point Process, the Class of Stochastic Differential Equations, and Their Applications

In this section, we overview how the physically motivated point process proposed in [21,22,23] was applied to model trading activity and absolute returns in the financial markets. We also discuss numerous extensions of the model into some related research topics, such as superstatistics and anomalous and non-homogeneous diffusion.

### 2.1. The Multiplicative Point Process Model

Let us consider signal $I\left(t\right)$ composed of pulses with profiles given by $A_{k}\left(x\right)$:(1)$$I\left(t\right)=\sum_{k}A_{k}\left(t-t_{k}\right),$$
where $t_{k}$ is the event (pulse) time. There are many physical and social systems, which generate signals of such nature: electric current [57], music [58], human heartbeat [59], internet traffic [32], or trading activity [29] to name a few.

As most profiles of the pulses are brief, it is trivial that they would influence only high frequencies corresponding to the typical inverse pulse length. If we are interested in longer-term dynamics, it is sufficient to assume that the Kronecker delta function well approximates the profile, $A_{k}\left(x\right)=a_{k}\delta \left(x\right)$. Many such systems are driven by the flow of identical or similar objects, such as electrons, packets, or trades. This lets us simplify (Equation 1) and investigate it as a temporal point process with unit events. Such a process can be either described by the event times $\left\{t_{k}\right\}$ or by the inter-event times $\left\{\tau_{k}=t_{k+1}-t_{k}\right\}$.

The inter-event times are a far more convenient choice to model as they at least can give a semblance of the stationarity, while event times are obviously non-stationary as $\left\{t_{k}\right\}$ is monotonically increasing series. In [21,22,23], it was analytically shown that a relatively slow autoregressive AR(1) Brownian motion of $\tau_{k}$ yield 1/f fluctuations of the signal I(t). The author of [29] has built upon this observation and introduced multiplicative point process for the inter-event time
(2)$$\tau_{k+1}=\tau_{k}+\sigma^{2}\gamma \tau_{k}^{2\mu -1}+\sigma \tau_{k}^{\mu }\varepsilon_{k}.$$

In the above, it is assumed that inter-event time fluctuates due to exogenous perturbations. Perturbations are assumed to be standard uncorrelated Gaussian random variables, $\varepsilon_{k}$. The general rate of change is governed by σ, while γ is the damping constant. Multiplicativity, specified by μ, ensures that $I\left(t\right)$ is multifractal and has a power–law PDF. This point process model has found its use for the analysis of 1/f noise and long-range memory in many diverse phenomena such as musical rhythm spectra [58], human cognition [60], human interaction dynamics [61], turbulence [62], and few others [63,64,65,66]. Inspired by this model, [67] has shown under which conditions $1/f^{\beta }$ spectrum can arise from reversible Markov chains.

After closer examination, it should be evident that Equation (Equation 2) can be seen as an iterative solution of a certain SDE if Euler–Maruyama method was used [68]. Hence the corresponding Langevin SDE can be trivially recovered from the iterative relation (Equation 2):(3)$$d\tau =\sigma^{2}\gamma \tau^{2\mu -1}dk+\sigma \tau^{\mu }dW_{k}.$$

Here *W* is uncorrelated standard Wiener process. Note that this SDE is in the event space (or *k*–space) and not in the real time. Further, this SDE must be solved by restricting the diffusion of the inter-event time τ to some arbitrary interval $\left[\tau_{\min},\tau_{\max}\right]$ on the positive half-plane as otherwise this SDE may not have a stationary distribution. If stationary distribution exists, then the stationary PDF of τ is a power–law:(4)$$p_{k}\left(\tau \right)=\frac{\alpha +1}{\tau_{\max}^{\alpha +1}-\tau_{\min}^{\alpha +1}}\tau^{\alpha },\;\alpha =2\left(\gamma -\mu \right).$$

Yet the main result of [29] is the power–law statistical properties of $I\left(t\right)$. In the limit $\tau_{\min}\to 0$ and $\tau_{\max}\to \infty$ PSD of $I\left(t\right)$ in arbitrarily long range of frequencies has a power–law slope:(5)$$S\left(f\right)∼1/f^{\beta },\;\beta =1+\frac{2\left(\gamma -\mu \right)}{3-2\mu }.$$

The number of events in a selected time window, for example number of trades per minute, also has a power–law distribution [29]:(6)$$p\left(N\right)∼N^{-2\left(\gamma -\mu \right)-3}.$$

Formally, one could define the number of events in a window of length *w* as $N\left[t\right]=\int_{t}^{t+w}I\left(u\right)du$ (here the square brackets indicate that *N* is in discrete time). These analytical results can be confirmed by numerical simulation (see Figure 1).

### 2.2. The Class of Non-Linear Stochastic Differential Equations

In [33,69,70,71], we have made a transition from *k*-space to real time and this enabled us to model trading activity and absolute returns in the financial markets not only qualitatively, but quantitatively, too. The transition from SDE in *k*-space, Equation (Equation 3), to real time is achieved by substitution dt=τdk, which yields:(7)$$d\tau =\sigma^{2}\gamma \tau^{2\mu -2}dt+\sigma \tau^{\mu -1/2}dW.$$

Modeling inter-event time in real time makes less sense than in the *k*-space, so let us change the variable to the number of events per unit time $x=\frac{1}{\tau }$. Applying Itô transformation yields:(8)$$dx=\sigma^{2}\left(\eta -\frac{\lambda }{2}\right)x^{2\eta -1}dt+\sigma x^{\eta }dW.$$

In the above, we have introduced a more convenient set of parameters:(9)$$\eta =\frac{5}{2}-\mu ,\;\lambda =2\left(\gamma -\mu \right)+3.$$

As far as SDE (Equation 8) corresponds to the point process defined by Equation (Equation 2), the results for stationary PDF and PSD should apply:(10)$$\begin{array}{cc} p\left(x\right) & ∼x^{-\lambda },\;S\left(f\right)∼1/f^{\beta },\;\beta =1+\frac{\lambda -3}{2\eta -2}. \end{array}$$

The validity of these theoretical predictions was extensively checked numerically (see Figure 2 for a quick example) and also, in [72], proven analytically. The analytical proof provided in [72] allows interpreting the process modeled by SDE (Equation 8) in a more general context. In fact we can model any process possessing these power–law statistical properties, even processes, which make less sense from the perspective of the original point process.

Equation (Equation 8) and similar random walk models have been used to model the EUR/CHF exchange rate [73]. It has also lead to numerous modifications by our group, which we discuss in detail in the following subsections.

### 2.3. Reproducing the Long-Range Memory Using GARCH(1,1) Process

Autoregressive conditional heteroscedasticity (abbr. ARCH) family models [74,75,76,77,78,79] are quite popular forecasting tools among professional traders as well as researchers interested in the long-range memory phenomenon. Unlike SDEs, ARCH family models have explicitly built-in memory, which is built-in either via explicit dependence on the numerous previous states, infinitely many in the case of the ARCH(∞) model [80,81,82], or via fractional integration procedure, which introduces memory similar to the one present in the fractional Brownian motion, as in the fractionally integrated GARCH (abbr. FIGARCH) model [83,84,85]. In [86], we have shown that it is possible to modify the GARCH(1,1) model, which is Markovian in nature, to reproduce 1/f spectrum.

Generalized autoregressive conditional heteroskedasticity (abbr. GARCH) processes can be approximated by the diffusion processes. There are two competing approaches, which yield continuous approximations of GARCH processes using sets of SDEs. One of the approaches was proposed by Nelson [87] and the other by Kluppelberg et al. [88,89]. In the GARCH(1,1), Nelson’s approach is easier to apply, but has a drawback that the resulting COGARCH(1,1) would be driven by two sources of noise, instead of the one in the GARCH(1,1). Yet, we can circumvent the problem by ignoring the observed heteroskedastic economic variable $z_{t}$ and focusing on the approximation of the volatility process, $\sigma_{t}^{2}$, of GARCH(1,1):(11)$$\begin{array}{cc} z_{t} & =\sigma_{t}\omega_{t}, \end{array}$$
(12)$$\begin{array}{cc} \sigma_{t}^{2} & =a+bz_{t-1}^{2}+c\sigma_{t-1}^{2}=a+b\sigma_{t-1}^{2}\omega_{t-1}^{2}+c\sigma_{t-1}^{2}. \end{array}$$

In the above, $\omega_{t}$ is the noise, while *a*, *b*, and *c* are the GARCH(1,1) model parameters. For Nelson’s approach to work, we need to compute first and second moments of change in volatility. With the usual GARCH(1,1) we obtain SDE for geometric Brownian motion [86].

Now lets introduce non-linearity into Equation (Equation 12). In [86], we have explored two such options:(13)$$\begin{array}{cc} \sigma_{t}^{2} & =a+b\sigma_{t-1}^{\mu }\omega_{t-1}^{\mu }+c\sigma_{t-1}^{2}, \end{array}$$
(14)$$\begin{array}{cc} \sigma_{t}^{2} & =a+b\sigma_{t-1}^{\mu }{\left|\omega_{t-1}\right|}^{\mu }+\sigma_{t-1}^{2}-c\sigma_{t-1}^{\mu }. \end{array}$$

Both of these options can be approximated by SDEs belonging to the class of SDEs (Equation 8) with λ=μ and η=μ/2. Consequently both of these options reproduce 1/f spectrum with μ=3. Other parameters, *a*, *b*, and *c*, influence only the additional terms, which restrict the diffusion of $\sigma_{t}^{2}$. Setting these values too high shrinks the interval and the power–law distribution becomes extremely hard to observe.

### 2.4. Anomalous Diffusion in the Long-Range Memory Process

SDE (Equation 8) can be also seen to describe a heterogeneous diffusion in a non-linear potential. Such diffusion leads to anomalous growth in variance [90]
(15)$$\langle {\left[x\left(t\right)-\langle x(t)\rangle \right]}^{2}\rangle ∼t^{\theta },\;\theta =\frac{1}{1-\eta }.$$

This phenomenon is also known as anomalous diffusion [91,92,93]. If θ=1 then the process exhibits normal diffusion. Otherwise if 0<θ<1, the diffusion is slower than normal and is referred to as sub-diffusion. The diffusion may also be faster, if 1<θ<2, in that case it is called super-diffusion.

The anomalous diffusion can be obtained from SDE (Equation 8) only for specific parameter values such as λ<1 and η<1/2 [90]. Because power–law slope of the PSD, β, varies between 0 and 2, from Equation (Equation 10), it follows that anomalous diffusion and power–law noise can be observed at the same time only for negative parameter η values, specifically for η<(λ−1)/2 and λ<1; however, for these parameters values numerical simulation would become very slow and inefficient [72]; therefore, we have considered generalizing SDE (Equation 8) by considering non-Gaussian white noise.

In [94], we have considered Lévy α-stable noise. SDE equivalent to SDE (Equation 8), but with Lévy α-stable noise takes the following form:(16)$$\frac{dx}{dt}=\gamma \left(\eta ,\lambda ,\alpha \right)x^{\alpha (\eta -1)+1}+x^{\eta }\xi_{\alpha }\left(t\right).$$

Here, $\xi_{\alpha }\left(t\right)$ is a white noise, the intensity of which is distributed according to the symmetric Lévy α-stable distribution. The characteristic function of the noise intensity is given by:(17)$$\langle \exp\left(ik\xi_{\alpha }\right)\rangle =\exp\left(-\sigma^{\alpha }{\left|k\right|}^{\alpha }\right).$$

Here, α is the index of stability and σ is the scale parameter. We interpret SDE (Equation 16) in an Itô sense and it can also be written in the form
(18)$$dx=\gamma \left(\eta ,\lambda ,\alpha \right)x^{\alpha (\eta -1)+1}dt+x^{\eta }dL_{t}^{\alpha }.$$

Here, $dL_{t}^{\alpha }$ stands for the increments of Lévy α-stable motion $L_{t}^{\alpha }$. If SDE (Equation 16) is solved with reflective boundary conditions and
(19)$$\gamma \left(\eta ,\lambda ,\alpha \right)=\frac{\sin\left[\pi \left(\frac{\alpha }{2}-\alpha \eta +\lambda \right)\right]}{\sin[\pi (\alpha (\eta -1)-\lambda )]}\frac{\Gamma (\alpha \eta -\lambda +1)}{\Gamma (\alpha (\eta -1)-\lambda +2)},$$

then generalized SDE (Equation 16) generate time series with power–law steady-state PDF and power–law PSD:(20)$$p\left(x\right)∼x^{-\lambda },\;S\left(f\right)∼\frac{1}{f^{\beta }},\;\beta =1+\frac{\lambda -3}{\alpha (\eta -1)}.$$

Extensive numerical simulations have shown that due to the presence of the multiplicative Lévy α-stable noise in Equation (Equation 16) both sub-diffusion and super-diffusion can be observed together with power–law noise even for positive η values [95]; however, no analytical expression for anomalous diffusion exponent dependence on SDE parameters has been derived yet.

In Figure 3, we show a sample series of the solutions of SDE (Equation 16) and the statistical properties of the series when the noise is Lévy α-stable noise with α=1. The other SDE (Equation 16) parameters were picked so 1/f spectrum would be reproduced. As can be seen in the subfigure (**a**), ongoing diffusion is disrupted by huge jumps, which are characteristic to Lévy flights.

If we consider modeling only sub-diffusive processes, then we can study another generalization of SDE (Equation 8), originally proposed in [96]. If we start with a Markovian process described by the Itô SDE
(21)$$dx\left(\tau \right)=f\left[x(\tau )\right]d\tau +g\left[x(\tau )\right]dW\left(\tau \right).$$

The drift and diffusion functions of the above SDE are given by
(22)$$f\left(x\right)=\sigma^{2}\left(\eta -\frac{\lambda }{2}\right)x^{2\eta -1},\;g\left(x\right)=\sigma x^{\eta }.$$

We interpret the time τ as an internal (operational) time. For the trapping processes that have a distribution of the trapping times with power–law tails, the physical time t=T(τ) is given by the strictly increasing $\alpha_{+}$-stable Lévy motion defined by the Laplace transform
(23)$$\langle e^{-kT(\tau )}\rangle =e^{-\tau k^{\alpha_{+}}}.$$

Here, the parameter $\alpha_{+}$ takes the values from the interval $0<\alpha_{+}<1$. Thus, the physical time *t* obeys the SDE
(24)$$dt\left(\tau \right)=dL^{\alpha_{+}}\left(\tau \right),$$
where $dL^{\alpha_{+}}\left(\tau \right)$ stands for the increments of the strictly increasing $\alpha_{+}$-stable Lévy motion $L^{\alpha_{+}}\left(\tau \right)$. For such physical time *t* the operational time τ is related to the physical time *t* via the inverse $\alpha_{+}$-stable subordinator
(25)S(t)=inf{τ:T(τ)>t}.

Such subordination leads to power spectral density
(26)$$S\left(f\right)∼\left\{\begin{array}{cc} \frac{1}{\omega^{\beta }}, & 1-\alpha_{+}<\beta <1+\alpha_{+},\\ \frac{1}{\omega^{1+\alpha_{+}}}, & \beta >1+\alpha_{+}. \end{array}\right.,\;\beta =1+\frac{\alpha_{+}\left(\lambda -3\right)}{\left(\eta -1\right)}$$

Proposed SDEs (Equation 8), (Equation 16), and (Equation 21) have served as a basis to study heterogeneous diffusion in a non-homogeneous medium [90,96,97] and time subordinated processes [98,99] as well as the effects of non-linear variable transformations [100,101].

In paper [98], we investigated the distinction between the internal time of the system and the physical time as a source of 1/f noise. We have introduced the internal (operational) time into the earlier point process [21,22,23] together with additional equations relating the internal time to the physical time. In this scenario, we can still recover power–law statistical features similar to the ones obtained by solving Equation (Equation 8). In the financial markets, the internal time could reflect the fluctuating human activity, e.g., trading activity, yielding the long-range correlations in the volatility. The effective approach for the solution of highly non-linear SDEs was proposed [98] by a suitable choice of the internal time and variable steps of integration.

The effects of non-linear variable transformations [100,101] suggest that long-range memory in certain cases can be just a measurement effect. As far as the non-linear transformation of the observable *x* to *y*
(27)$$x=\frac{1}{y^{\delta }},$$
with δ being the transformation exponent, yields SDE for the variable *y* of the same form such as Equation (Equation 8) for *x*.

### 2.5. Inverse Cubic Law for Long-Range Correlated Processes

The inverse cubic law is an established stylized fact stating that the cumulative distributions of various financial market time series such as the number of trades, the trade volume, or the return [12,14,15,19]. Thus, this law is as important for the modeling as the consideration of long-range memory and fractal scaling, which are also stylized facts [6,12,14,15,19]. We have in proposed [102] that the non-linear SDE yields both the power–law behavior of the PSD and the inverse cubic law of the cumulative distribution. This was achieved using the idea that when the market evolves from calm to violent behavior there is a decrease of the delay time of multiplicative feedback of the system in comparison to the driving noise correlation time. This results in a transition from the Itô to the Stratonovich sense of the SDE and yields a long-range memory process.

We start from a simple quadratic SDE
(28)$$dx=x^{2}\circ_{\alpha }dW$$
where α is the interpretation parameter, defining the α-dependent stochastic integral of the SDE (Equation 28),
(29)$$\int_{0}^{T}f\left(x\left(t\right)\right)\circ_{\alpha }dW_{t}\equiv \lim_{N\to \infty }\sum_{n=0}^{N-1}f\left(x\left(t_{n}\right)\right)\Delta W_{t_{n}}.$$

Here, $t_{n}=\frac{n+\alpha }{N}T$ with 0≤α≤1. Natural choices of the parameter α are: (i) α=0, pre-point (Itô convention), (ii) α=1/2, mid-point (Stratonovich convention), and (iii) α=1, post-point (Hänggi–Klimontovich, kinetic, or isothermal convention) [103].

The quadratic SDE (Equation 28) is the simplest multiplicative SDE without the drift term symmetric for the positive and negative deviations of some observable *x*. More generally, the same process can be described by the delayed SDE [103]
(30)$$dx\left(t\right)=f\left(x(t)\right)dt+g\left(x(t-\delta )\right)\zeta_{t}^{\tau }dt.$$

Here, $f\left(x\right)$ represents arbitrary deterministic drift of the observable *x*, while $g\left(x\right)$ effectively controls the diffusion as $\zeta_{t}^{\tau }$ is the noise term, which is assumed to have correlation time τ. Note that the diffusion function depends on the delayed value of the observable *x* (by time interval δ).

It may be shown [103] that in the limit δ→0 and τ→0 (under the condition δ/τ=const) SDE (Equation 30) can be transformed into
(31)$$dx=f\left(x(t)\right)dt+g\left(x(t)\right)\circ_{\alpha }dW$$
with the interpretation parameter being determined by
(32)$$\alpha \left(\frac{\delta }{\tau }\right)≃\frac{1}{2\left(1+\delta /\tau \right)}.$$

Under the perturbation by the white noise, in a case of τ≪δ, even for a short delay in feedback δ, we achieve the Itô outcome, because there is no correlation between the sign of the noise $\zeta_{t}$ and the time-derivative of the feedback $g\left(x\right)$. On the contrary, under the perturbation by the correlated noise, τ≫δ, a correlation emerges between the sign of $\zeta_{t}$ and the time-derivative of $g\left(x\right)$. In this case the correlation yields the Stratonovich outcome [103].

In general, the value of α may depend on the coordinate *x* and/or other system’ parameters. SDE (Equation 28) with α≠0 may be transformed into SDE in Itô sense
(33)$$dx=2\alpha x^{3}dt+x^{2}dW.$$

This SDE is a particular case of the general Itô Equation (Equation 8) yielding the power–law steady-state PDF and the power–law PSD (Equation 10). These SDEs become identical for η=2 and $\lambda =4\left(1-\alpha \right)$.

Let us note that $1/f^{\beta }$ noise emerges due to the large fluctuations in the time series, while the finite time studies reveal the commonly observed magnitudes of the observable. The common fluctuations can be modeled by the familiar in the financial application’s Itô SDEs. On the other hand, the large rapid fluctuations of the violent market arise due to the strong correlated influences; the processes of such a market are fast, all durations become short in comparison to the herding correlation time, and, consequently, the market should be modeled by the Stratonovich version of SDE.

For the modeling of such dynamics, we generalize Equations (Equation 28) and (Equation 33) with *x*-dependent parameter α(x). Let
(34)$$dx=2\alpha \left(x\right)x^{3}dt+x^{2}dW,$$
with, e.g.,
(35)$$\alpha \left(x\right)=\frac{1}{2}\left[1-\exp\left\{-{\left(\frac{x}{x_{c}}\right)}^{2}\right\}\right],$$
where $x_{c}$ is the Itô to Stratonovich interpretations crossover parameter. Equations (Equation 34) and (Equation 35) represent transition from Itô to Stratonovich convention with an increase in the variable *x* and decrease of the delay time of multiplicative feedback for larger *x*, according to the Wong–Zakai theorem [103]. Detailed numerical analysis of the model represented by Equations (Equation 34) and (Equation 35) is presented in paper [102].

### 2.6. $1/f^{\beta }$ Noise with Distributions Other Than Power–Law

Solutions of the SDE (Equation 8) will always have power–law statistical properties of the (Equation 10) form; however, often noise with $1/f^{\beta }$ PSD is distributed according to PDF, which is not power–law, but Gaussian or some other distribution. Here, we review two different approaches, which allow for other distributions to be observed in time series with $1/f^{\beta }$ spectrum: superstatistical and coupled SDE approaches.

In [104], it was suggested that the Poissonian-like process with the slowly changing average inter-event time may be represented as the superstatistical process exhibiting 1/f noise. It was assumed that the inter-event time $\tau_{k}$, obtained by solving Equation (Equation 2), represents not the actual (observed) inter-event time, but its average (reciprocal of the event rate). In this setup, the actual inter-event time ${\hat{\tau }}_{k}$ would be given by the conditional probability
(36)$$\varphi \left({\hat{\tau }}_{k}|\tau_{k}\right)=\frac{1}{\tau_{k}}e^{-{\hat{\tau }}_{k}/\tau_{k}},$$
similar to the non-homogeneous Poisson process. This additional randomization has no influence on the lower frequencies of the PSD and the intensity of the signal.

The PDF of the observed inter-event time ${\hat{\tau }}_{k}$ may be derived from the superstatistical model,
(37)$$p\left({\hat{\tau }}_{k}\right)=\int_{0}^{\infty }\varphi \left({\hat{\tau }}_{k}|\tau_{k}\right)p_{k}\left(\tau_{k}\right)d\tau_{k}.$$

Equations (Equation 36) and (Equation 37) generate the *q*-exponential distribution used in the non-extensive statistical mechanics and many real systems [105]. Detailed analytical derivations and the numerical verification were presented in [104].

In the paper [38], a similar superstatistical approach was taken with respect to the intensity of the signal *x*, obtained by solving SDE (Equation 8). The observed series $\hat{x}$ is assumed to be generated from *x* series by applying exogenous noise, which is described by an arbitrary conditional distribution $\varphi (\hat{x}|x)$. In this approach, the steady-state distribution of $\hat{x}$ is given by
(38)$$p\left(\hat{x}\right)=\int_{0}^{\infty }\varphi \left(\hat{x}|x\right)p\left(x\right)dx.$$

Analytical and numerical analysis of inter-trade duration, the trading activity, and the return using the superstatistical method with the exponential and normal distributions of the local signal, driven by the stochastic process, were discussed in detail in [38].

In later sections of this paper, we show that the superstatistical approach is not the only approach that allows us to change the observed signal PDF. The coupled SDE approach, proposed in [99], allows for more flexibility and easier interpretation of how the statistical properties become independent of each other. The general form of the set of coupled SDEs was derived from the scaling properties needed for the realization of $1/f^{\beta }$ noise [99]
(39)$$\begin{array}{cc} dx & =f\left(x\right)y^{2\eta }dt+g\left(x\right)y^{\eta }dW_{1}, \end{array}$$
(40)$$\begin{array}{cc} dy & =\sigma^{2}\left(\eta +1-\frac{\lambda }{2}\right)y^{2\eta +1}dt+\sigma y_{t}^{\eta +1}dW_{2}. \end{array}$$

Here, f(x) and g(x) are arbitrary drift and diffusion functions, which determine the stationary PDF of *x*; $W_{1}$ and $W_{2}$ are uncorrelated standard Wiener processes. The first equation describes the changes in the intensity of the signal, while the second equation represents fluctuations in the rate of change. These coupled SDEs allow for $1/f^{\beta }$ spectrum to be reproduced together with arbitrary steady-state PDF of the observed value *x*. It was shown that the power–law slope of the PSD, β, of the time series of *x* generated by solving SDEs (Equation 39) and (40) depends on the parameters η and λ as follows
(41)$$\beta =1+\frac{\lambda -1}{2\eta }.$$

In Figure 4, we show that one can obtain a Gaussian distribution of *x* (subfigure (**b**)) together with 1/f spectrum (subfigure (**c**)). In subfigure (**a**), one can visually see the impact of the variations in the rate of change.

### 2.7. Reproducing Statistical Properties of the Financial Markets

While qualitatively, the trading activity and the absolute returns have power–law distributions and exhibit long-range memory property [14,19], corresponding empirical statistical properties have a finer structure. In order to reproduce the empirical statistical properties in detail, some modifications to the SDE are needed.

The author of [13] has determined that Hurst exponents of the trading activity time series of 1000 US stocks are remarkably close: H≈0.85. This implies that the PSD of the trading activity should have a power–law slope β=2H−1≈0.7. The author of [13] has also discovered the that slope of the PDFs of the trading activity also has a power–law tail with exponent λ≈4.4. It would be impossible to reproduce such values by using SDE (Equation 8), because Equation (Equation 10) implies that if λ>3, then β>1. In our analysis of 26 US stocks [106], we have confirmed the slope of the PDF, but we have observed a more complicated PSD, with two slopes instead of one (β<1 for both slopes).

Both of these issues are resolved by a modified SDE for trade intensity, *n* [33]:(42)$$dn=\sigma^{2}\left[\eta -\frac{\lambda }{2}+{\left(\frac{n_{0}}{n}\right)}^{2}\right]\frac{n^{2\eta -1}}{{\left(n\epsilon +1\right)}^{2}}dt+\sigma \frac{n^{\eta }}{n\epsilon +1}dW.$$

The problem of the two PSD slopes is resolved, because this SDE has two different effective η values. For $n\gg \epsilon^{-1}$ the effective η is equal to the specified parameter value (in the numerical simulations we have used η=5/2, thus ${\hat{\eta }}_{1}=5/2$). For $n\ll \epsilon^{-1}$ the effective η is one smaller than the specified parameter value ${\hat{\eta }}_{2}=\eta -1=3/2$). The slope of the PDF increases from the value predicted in Equation (Equation 10) due to integration, as trading activity is defined as number of trades per time window *w*, or in the current parametrization, an integral of trade intensity: $N\left[t\right]=\int_{t}^{t+w}n\left(u\right)du$.

Parameter $n_{0}$ and the related term in the drift function ensure that *n* would not become very small as the term causes the potential to rapidly grow for $n<n_{0}$. This helps us avoid negative trade intensities, which are impossible by definition, as well as ensure some level of minimal trading activity, which in our experience may differ for different stocks and different markets [37,106].

In Figure 5, we have shown that the stochastic model can match statistical properties of MMM stock traded on NYSE. While the matches are not perfect, some of the noticeable differences can be explained by the fact that the stochastic model does not take into account intraday seasonalities.

Reproducing statistics of absolute return requires another modification of the SDE [36]. Our empirical analysis, confirmed by the other authors [105], indicated that the *q*-Gaussian distribution [38,107] seems to be a good fit for the empirical absolute return, defined as the log–price difference, distribution. This is achieved by:(43)$$dx=\sigma^{2}\left[\eta -\frac{\lambda }{2}-{\left(\frac{x}{x_{\max}}\right)}^{2}\right]\frac{{\left(1+x^{2}\right)}^{\eta -1}}{{\left(1+\epsilon \sqrt{1+x^{2}}\right)}^{2}}xdt+\sigma \frac{{\left(1+x^{2}\right)}^{\frac{\eta }{2}}}{1+\epsilon \sqrt{1+x^{2}}}dW.$$

To reproduce the full complexity of the empirical data, another ingredient is needed, namely external noise, which can be understood as an effect of news flow or the distortions caused by the discrete order flow:(44)$$r_{t}=\xi \left\{r_{0}=1+\frac{2}{w}\left|\int_{t-w}^{t}x\left(u\right)du\right|,q=1+2/\lambda_{2}\right\}.$$

This relation was inspired by the superstatistical approach (discussed in Section 2.6) and determined by trying to fit the empirical data as best we can. We have empirically determined that the best fit is obtained when ξ is a process that generates uncorrelated random variates from a *q*-Gaussian distribution with q≈1.4 ($\lambda_{2}\approx 5$) and $r_{0}$ being one minute (w≈60s) moving average filter of the solutions of SDE (Equation 43). Using this model, we were able to reproduce empirical statistical properties of stock from New York (abbr. NYSE) and Vilnius stock exchanges (abbr. VSE) [36,37].

In Figure 6, we have demonstrated that the stochastic model reproduces empirical data reasonably well from NYSE and VSE. Some of the noticeable differences can be observed because we do not take into account the intraday seasonality, and we do not directly take into account that VSE had relatively low liquidity (many one minute time intervals have zero returns). Differing liquidity is a likely explanation for the differences seen between NYSE and VSE, too.

### 2.8. Variable Step Method for Solving Non-Linear Stochastic Differential Equations

Note that SDEs (Equation 8), (Equation 42), and (Equation 43) are not Lipshitz continuous [68]; thus, they have to be solved by imposing boundary conditions, which would prevent the explosion of the solutions. An alternative way to achieve Lipshitz continuity is to include additional terms for restricting diffusion, which would have no detrimental effects on the PSD and PDF of the time series. Such is the role of the $n_{0}$ term in SDE (Equation 42) and $x_{\max}$ term in SDE (Equation 43).

Lacking Lipshitiz continuity causes another complication in solving the SDEs: the standard Euler–Maruyama or Milsten methods [68] do not yield good results with reasonable step sizes. This complication is resolved by using a variable step size. The core idea is to use a larger step size whenever the anticipated changes would be small and use the smaller step size whenever significant changes are coming. The mathematical form of the variable step size is often unique to the SDE being solved, but a good rule of thumb would be to linearize the drift and the diffusion functions. See [69,70] for more details.

For example, SDE (Equation 8) in our works is solved by the following set of difference equations:(45)$$\begin{array}{cc} x_{i+1} & =x_{i}+\kappa^{2}\left(\eta -\frac{\lambda }{2}\right)x_{i}+\kappa x_{i}\varepsilon_{i}, \end{array}$$(46)$$\begin{array}{cc} t_{i+1} & =t_{i}+\kappa^{2}x^{2-2\eta }. \end{array}$$

In the above κ is a small number that acts as an error tolerance parameter. The smaller it becomes, the better $x_{i}$ reproduces desired statistical properties given by Equation (Equation 10), but at the expense of numerical computation time.

Similarly, this variable step method can be also applied to SDEs with α-stable Lévy noise. For example, we can solve SDE (Equation 16) numerically by using the following set of difference equations
(47)$$\begin{array}{cc} x_{k+1} & =x_{k}+\kappa^{\alpha }\gamma x_{k}+\frac{\kappa }{\sigma }x_{k}\xi_{k}^{\alpha }, \end{array}$$
(48)$$\begin{array}{cc} t_{k+1} & =t_{k}+\frac{\kappa^{\alpha }}{\sigma^{\alpha }}x_{k}^{-\alpha (\eta -1)}, \end{array}$$
where $\xi_{k}^{\alpha }$ is a random variable having α-stable Lévy distribution. This set of difference equations should be solved only with the reflective boundaries at $x=x_{\min}$ and $x=x_{\max}$ using the projection method [108]. In nutshell, if the variable $x_{k+1}$ acquires the value outside of the interval $\left[x_{\min},x_{\max}\right]$ then the value of the nearest reflective boundary is assigned to $x_{k+1}$. Iterative equations for SDEs (Equation 42) and (Equation 43) are a bit more complicated [36,106], but they still remain qualitatively the same.

Note that the introduction of the variable time step into the numerical solution of an SDE is equivalent to introducing the subordination scheme directly into the SDE, when internal time and physical time are related by a non-linear transformation [98].

## 3. Agent-Based Model of the Long-Range Memory in the Financial Markets

In the previous section, we have discussed how our group has started from the physically motivated point process model and arrived at the general class of SDEs reproducing long-range memory phenomenon; however, this generality has its drawback: microscopic mechanisms of the modeled systems are ignored. We then tried to investigate some existing financial ABMs for the possibility to derive SDE of a similar form to SDE (Equation 8). We have failed to do so with some prominent yet complicated ABMs, such as the ones proposed in [109,110] (for more prominent ABMs of the time, which include some other candidates we have tried, see [111]); however, we have found success with Kirman’s herding model, initially proposed in [112] and later analyzed in financial market context by [113,114].

### 3.1. Kirman’s Herding Model

Kirman’s herding model can be defined via two one-step transition probabilities in a system with two possible states:(49)$$\begin{array}{cc} p\left(X\to X+1\right) & =\left(N-X\right)\left[\sigma_{1}+hX\right]\Delta t, \end{array}$$(50)$$\begin{array}{cc} p\left(X\to X-1\right) & =X\left[\sigma_{2}+h\left(N-X\right)\right]\Delta t. \end{array}$$

In the above, *X* is the number of agents in state 1 and *N* is the total number of agents within the system. Total number of agents is conserved, so the number of agents in state 2 is trivially given by N−X. Here, Δt is a short time window during which only one transition should be likely. Transitions may occur either due to independent behavior (governed by parameters $\sigma_{i}$), or due to recruitment (governed by parameter *h*). Using birth–death process formalism [115] it is easy to find SDE corresponding to Kirman’s herding model with x=X/N:(51)$$dx=\left[\left(1-x\right)\sigma_{1}-x\sigma_{2}\right]dt+\sqrt{2hx\left(1-x\right)}dW.$$

### 3.2. Kirman’s Herding Model for the Financial Markets

Evidently, SDE (Equation 51) is not of the same form as SDE (Equation 8), but we have not yet discussed the meaning of states 1 and 2. In many financial ABMs of the time, it was a common choice to assume that agents represent chartist and fundamentalist traders [111]. Assuming that chartist traders trade based on the wide variety of technical trading tools, which often produce conflicting predictions, their excess demand (difference between the supply and demand generated by the group as a whole) is given by:(52)$$D_{c}=r_{0}X_{c}\left(t\right)\xi \left(t\right),$$
where $X_{c}\left(t\right)$ is the number of chartist traders and $\xi \left(t\right)$ is their average mood (describing average sentiment to buy or sell). The relative impact of the chartists’ traders in comparison to fundamentalist traders is given by $r_{0}$. Fundamentalist traders on the other hand are often assumed to trade based on the quantity known as a fundamental price, $P_{f}$, with the expectation that the price, $P\left(t\right)$, in the long run, will converge towards the fundamental price. Under this assumption, their excess demand is given by:(53)$$D_{f}=X_{f}\left(t\right)\ln\frac{P_{f}}{P\left(t\right)}.$$

Using the excess demand functions of the both groups, we can use Walras law [116] to obtain the expression for the price [40,113]:(54)$$P\left(t\right)=P_{f}\exp\left[r_{0}\frac{X_{c}\left(t\right)}{X_{f}\left(t\right)}\xi \left(t\right)\right].$$

The log–return of the price is evidently given by:(55)$$r_{w}\left(t\right)=\ln P\left(t\right)-\ln P\left(t-w\right)=r_{0}\frac{x_{c}\left(t\right)}{x_{f}\left(t\right)}\zeta_{w}\left(t\right).$$

In the above, $\zeta_{w}\left(t\right)$ is the mood change function over time window *w*. As the mood changes on a very short time scale and we are interested in the long-term dynamics, we can simply assume that $\zeta_{w}\left(t\right)$ is some kind of uncorrelated noise and consider only a more slowly varying ratio between fractions of chartists and fundamentalists. As the total number of agents is fixed, we can define long-term component of return, modulating return, as:(56)$$y\left(t\right)=\frac{x\left(t\right)}{1-x\left(t\right)}.$$

SDE for the modulating return is given by:(57)$$dy=\left[\sigma_{1}+\left(2-\sigma_{2}\right)y\right]\left(1+y\right)dt+\sqrt{2hy}\left(1+y\right)dW,$$
which is roughly similar to the SDE (Equation 8) with η=3/2 and $\lambda =\frac{\sigma_{2}}{h}+1$.

This SDE can be generalized by introducing variable event rate $\tau \left(y\right)=y^{-\alpha }$. This addition can be explained by the fact that it is well known that returns and trading volume correlate and the best correlation is achieved between squared returns and volume [16,17,18,117], hence suggesting that α=2 is a likely candidate. With this extension and when considering only the highest powers of *y* (as the large *y* tend to influence the PSD), we obtain [40]:(58)$$dy=h\left(2-\sigma_{2}\right)y^{2+\alpha }dt+\sqrt{2hy^{3+\alpha }}dW.$$

Now this SDE is completely equivalent to the SDE (Equation 8) with $\eta =\frac{3+\alpha }{2}$ and $\lambda =\frac{\sigma_{2}}{h}+\alpha +1$. Consequently PSD of *y* will have a frequency range in which:(59)$$S_{y}\left(f\right)∼1/f^{\beta },\;\beta =1+\frac{\frac{\sigma_{2}}{h}+\alpha -2}{1+\alpha }.$$

In the later papers, we modified this herding ABM until it was able to reproduce the absolute return PDF and PSD close to the empirical absolute return PDFs and PSDs. In [118], we have shown that considering mood dynamics can help in reproducing fractured PSD. In [41], we have reliably introduced the exogenous noise, much similar to what was achieved with the SDE driven model in [36], into this ABM, thus producing a consentaneous model. In [119,120], we have explored the opportunities to control the fluctuations in the artificial financial markets driven by the herding ABM, showing that the random trading, control strategy suggested in [121], may also destabilize the market. In [42], we have removed the assumption about the exogenous noise and replaced it with order book dynamics, thus presenting another possible explanation for fracture in the PSD: it also arises due to market price lagging behind the changes in the equilibrium price, Equation (Equation 54). Notably, the order book version of the model was able to reproduce both trading activity and absolute return statistical properties at the same time.

In Figure 7, we have reproduced one of the figures from [41] to show how well the ABM can reproduce the empirical data from New York, Vilnius, and Warsaw stock exchanges (abbr. WSE). Here, we have shown that the model was able to reproduce 10 min absolute return PDFs and PSDs from the different stock exchanges, but in the original article, more intraday time scales are covered, and seasonality was also taken into account.

### 3.3. Kirman’s Herding Model, Voter Model, and the Opinion Dynamics Context

Attentive reader with a background in opinion dynamics will likely notice that Kirman’s model is remarkably similar to the well-known voter model [47,48,49]. They are identical, which has prompted us to question whether the voter model is truly a model for voters, which Fernandez–Garcia et al. in [122] also raised. This has lead us to explore and model statistical properties of spatially heterogeneous electoral data [43]. As we have noticed segregation effects in the electoral data, we have continued our investigation by considering the migratory nature of census and electoral data [44]. Similar approaches were taken by others as well. Sano and Mori [123] have looked into spatiotemporal Japanese election data in their model, assuming a noticeable fraction of stubborn voters who do not allow for the party’s popularity to drop below a certain threshold. Braha et al. [124] have considered spatiotemporal US election data and have also emphasized the role of opinion leaders and spatial variability of external influences. Fenner et al. [125,126] have started from a generative model inspired by survival analysis, but in later works transition to the SDE framework [127,128]. Michaud and Szilva [129] have fixed issues with the model originally proposed by Fernandez–Garcia et al. [122], mainly, they have redefined how the noise term is handled so that the model would be more mathematically well-posed. Marmani et al. [130] have provided a similar empirical analysis of Italian electoral data and provided an additional perspective from the point of view of Shannon entropy.

As is common in opinion dynamics [47,48,49], we have also explored the influence of network topologies on the statistical properties of Kirman’s herding model. Namely, we have demonstrated [131] a continuous transition from extensive case, characterized by localized interactions, Gaussian distributions, and Boltzmann entropy, to a non-extensive case, characterized by global interactions, *q*-Gaussian distribution, and Tsallis entropy. Similar results were demonstrated earlier by Alfarano and Milakovic [132], who have explored how Kirman’s herding model works on random, Barabasi–Albert, and small-world network topologies. Similar observations were also made in [133], but Carro et al. have used the so-called annealed approximation, which takes into account network structures better than the usual mean-field approximation.

Recently, we have also used the noisy voter model to model parliamentary presence [45]. A paper by Vieira et al. [134] has inspired us to look into the Lithuanian parliamentary presence data. Unlike Vieira et al., we have observed not a ballistic diffusion regime but superdiffusive behavior; however, both of these regimes can be obtained from the noisy voter model with imperfectly acting agents. Namely, agents can internally intend to attend the parliamentary session or skip, but the action itself may be random despite being conditioned on the intended action. As Vieira et al. have used fractional diffusion equation as a model, this result implies that it may be possible to fake long-range memory encoded in the fractional diffusion equation by using Markov models employing non-linear transformations of the voter model [101].

The classical voter model incorporates only a recruitment mechanism, despite other responses to social interaction being possible. For example, diamond model [135] posits that independence and anti-conformity mechanism may be important to understanding human social behaviors. Similarly, Latane social impact theory [136] predicts the importance of supportive interactions—namely, individuals strengthening the conviction of their like-minded peers. While this theory was recently studied in the opinion dynamics context [137,138], it has not been combined with the voter model. One could also consider majority-vote models [139,140,141] and q-voter models [142,143] as implementing some kind of support by the like-minded agents. In majority-vote models, recruitment is only possible if a majority of agents have opposing opinions (therefore, the majority becomes harder to convince, but the minority remains as susceptible to change). In most q-voter models, a group of *q* agents must share an opinion to convince a single agent. We have implemented supportive interactions by decreasing the transitions rates of the agents by an amount proportional to the number of like-minded agents. In some cases, these modifications cause the transition rates go to zero, which freezes the system state. Similar qualitative behavior is observed in works, which consider non-Markovian mechanisms, such as implicit opinion freezing or aging [144,145,146,147]. This serves as another example that highly non-linear Markovian models can lead to similar dynamics as the dynamics generated by the non-Markovian models.

## 4. Searching for the True Long-Range Memory Test

We have reviewed our experience of modeling long-range memory phenomena using Markovian models in the earlier sections. We have shown numerous examples of non-linearity causing behaviors and dynamics reminiscent of the models with true long-range memory (such as delayed feedback, aging, freezing, and fractional dynamics). In this section, we present our latest endeavor to find a statistical test, which would distinguish whether the real-life systems possess true or spurious long-range memory. We proposed a test earlier, based on the specific first-passage times, which we refer to as the burst and interburst duration analysis (abbr. BDA) [148,149,150,151].

Investigating empirical PDF of burst and interburst duration compared with the model properties, we have interpreted the observed long-range memory in the financial markets by ordinary non-linear SDEs representing multifractal stochastic processes with non-stationary increments [152,153]. One has to take into account the interplay of endogenous and exogenous fluctuations in the financial markets to build a comprehensive model of this complex system [154]. Non-linear SDEs might be applicable in the modeling of other social systems, where models of opinion or population dynamics lead to the macroscopic description by these equations [148,149,150,151]. The description by SDEs is an alternative to the modeling incorporating fractional dynamics, if power–law statistical properties are observed in the empirical data.

The BDA employs the dependence of first-passage time PDF on Hurst exponent *H* for the fractional Brownian motion [56,152,153,155].

FBM, FLSM, and ARFIMA [156,157,158] form the theoretical background of long-range memory and self-similar processes. These processes, first of all, served for the modeling of systems with anomalous diffusion and expected fractional dynamics [159]. We can consider fractional models possessing true long-range memory as they have correlated increments. Self-similar processes with non-Gaussian stable increments are essential for the modeling of social systems as well. In the financial markets, power–law distributions of noise often interplay with autocorrelations [160,161,162]. In [163], we implemented BDA for the order disbalance time series seeking to confirm or reject the long-range memory in the order flow. Further, we analyzed the same LOBSTER data of order flow in the financial markets [164] from the perspective of FLSM and ARFIMA models seeking to identify the impact of increment distributions and correlations on estimated parameters of self-similarity [165]. The revealed peculiarities of non-Gaussian fractional dynamics in this financial system raise new questions about whether used sample estimators are reliable. In this section, we test various long-range memory estimators such as mean squared displacement, absolute value estimator, Higuchi’s method, and BDA on discrete fractional Lèvy stable motion represented by the ARFIMA sample series.

### 4.1. Fractional Processes with Non-Gaussian Noise

FBM serves as a model of the correlated time series with stationary Gaussian increments and generalizes the classical Brownian motion [1]. One can define FBM, $B_{H}\left(t\right)$, of the index *H* (Hurst parameter) in the interval 0<H<1 as the Itô integration over classical Brownian motion *B*
(60)$$B_{H}\left(t\right)=\int_{-\infty }^{\infty }\left({\left(t-u\right)}_{+}^{d}-{\left(-u\right)}_{+}^{d}\right)dB\left(u\right),$$
where d=H−1/2, ${\left(x\right)}_{+}=\max\left(x,0\right)$. The parameter *H* in FBM quantifies fractal behavior, long-range memory, and anomalous diffusion. This is not the case for the other more general stochastic processes. Thus, in this contribution the Hurst parameter *H* is responsible only for the fractal properties of the trajectories. We will consider fractional Lèvy stable motion as more general process with non-Gaussian distribution $L_{H}^{\alpha }\left(t\right)$ representing an integrated process of independent and stable stationary increments $dL^{\alpha }\left(u\right)$ [156]
(61)$$L_{H}^{\alpha }\left(t\right)=\int_{-\infty }^{\infty }\left({\left(t-u\right)}_{+}^{d}-{\left(-u\right)}_{+}^{d}\right)dL^{\alpha }\left(u\right),$$
where parameter *d* depends on *H* and parameter of stable distribution α, d=H−1/α. The parameter α characterizes special class of stable, invariant under summation, distributions [166], useful in the modeling both super and sub-diffusion [159]. Here, we are interested in the symmetric zero mean, stable distribution defined by the stability index in the region 0<α<2. This new parameter is responsible for the power–law tails of the new PDF $P\left(x\right)∼{\left|x\right|}^{-1-\alpha }$.

FBM and FLSM exhibit identical self-similar scaling behavior in statistical sense,
(62)$$B_{H}\left(ct\right)∼c^{H}B_{H}\left(t\right),\;L_{H}^{\alpha }\left(ct\right)∼c^{H}L_{H}^{\alpha }\left(t\right),$$
where x∼y means that *x* and *y* have identical distributions. One can establish the relation with the fractal dimension of trajectories D=2−H [167]. In analogy to the notions used in fractal geometry, these types of processes can be considered self-similar.

Mean squared displacement (abbr. MSD) is another important statistical property of various complex systems. Mathematically it was introduced as an ensemble average of the possible microscopic trajectories x(t) [159]
(63)$$\langle {\left(x(t)-x(0)\right)}^{2}\rangle ∼t^{\lambda },\;\lambda =2d+1.$$

Note that Equation (Equation 63) is valid for the FBM, while the ensemble average of FLSM diverges [156]. For the FBM d=H−1/2, while for the FLSM λ is not defined. When d<0, one observes dynamics as sub-diffusion and for d>0 as super-diffusion.

In experimental or empirical data analysis, one usually deals with discrete-time sample data series $\left\{X_{i}\right\}$. It is challenging to decide which model to apply in the description of empirical data when diffusion is anomalous d≠0, as observed dynamics in the sample data can originate from the long-range memory or power–law of the noise. We will use the sample MSD defined as
(64)$$M_{N}\left(k\right)=\frac{1}{N-k+1}\sum_{i=0}^{N-k}{\left(X_{i+k}-X_{k}\right)}^{2}.$$

Let us also introduce increment process $\left\{Y_{i}=X_{i}-X_{i-1}\right\}$, which is extracted from the sample data series. In the case of the FBM increment process, it is called fractional Gaussian noise (abbr. FGN), and in the case of FLSM, it is called fractional Lèvy stable noise (abbr. FLSN). The authors in [156] provide evidence of FLSM non-ergodicity and that $M_{N}\left(k\right)∼k^{\lambda }$, where λ=2d+1, for large *N*, *k*, and N/k. Thus, the MSD sample analysis of time series with FLSM assumption becomes very important providing estimation of the memory parameter *d*. The long-range memory usually is defined through the divergence of autocovariance ρ(k), $\sum_{k=1}^{\infty }\rho \left(k\right)=\infty$, [11]
(65)$$\begin{array}{c} \rho \left(k\right)=\frac{1}{N-k+1}\sum_{i=1}^{N-k+1}Y_{i}Y_{i+k}=2^{-1}\{{\left(k+1\right)}^{2H}-2k^{2H}{+|k-1|}^{2H}\}\\ ∼H\left(2H-1\right)k^{-\gamma },\;k\to \infty . \end{array}$$

For the FGN, the exponent of autocorrelation is defined by the Hurst parameter γ=2−2H. We see that FBM is an essential long-range memory process with various statistical properties defined by the Hurst parameter. Thus, researchers use an extensive choice of statistical estimators to determine *H* and evaluate memory effects even when investigated time series deviate from the Gaussian distribution.

Accepting a more general FLSM approach, one has to reevaluate previously used estimators [163], as we now have more independent parameters. The stability index 0<α<2 and the memory parameter *d* both contribute to the observed sample properties. Since in the Lèvy stable case, the second moment is infinite the measure of noise autocorrelation, e.g., the co-difference [166,168], is used instead of covariance
(66)$$\tau \left(k\right)=∼k^{-(\alpha -\alpha H)}.$$

Note that the parameter γ=α−αH=α−αd−1, has a strong dependency on α, when for the Gaussian processes, it was considered just as the indicator of long-range memory. Consequently, the previously used sample power spectral density analysis, the rescaled range analysis [169,170,171], or multifractal detrended fluctuation analysis [172,173] has to be reevaluated from the perspective of FLSM [163,165].

Earlier, we have introduced the burst and interburst duration analysis (BDA) as one more method to quantify the long-range memory through the evaluation of *H* [149,152,153,163]. For the one dimensional bounded sample time series, any threshold divides these series into a sequence of burst $T_{j}^{b}$ and interburst $T_{j}^{i}$ duration, $j=1,\ldots ,N_{b}$. The notion of burst and interburst duration follows from the threshold first-passage problem initiated at the nearest vicinity of the threshold. The burst duration is the first-passage time from above and interburst from below the threshold, see [149,152,153,163] for more details. The empirical (sample) PDF (histogram) of $T_{j}$ gives us the information about *H*, as the power–law part of this PDF should be $T^{2-H}$ [56]. We have to revise the method of BDA from the more general perspective of FLSM [165], as the question of which properties can be recovered using this method is open and has to be investigated.

The method of absolute value estimator (abbr. AVE) works correctly even for the time series with infinite variance [11,167,168,174]. The method is based on mean value $\delta_{n}$ calculated from sample series $Y_{i}$ and evaluating its scaling with length of sub-series *n*. Divide the increment series $Y_{i}$ into blocks of size *n*, so that m·n=N, and average within each block to obtain the aggregated series $Y_{j}^{\left(n\right)}=\frac{1}{n}\sum_{i=(j-1)n+1}^{jn}Y_{i}$. Calculate $\delta_{n}$
(67)$$\delta_{n}=\frac{1}{m}\sum_{j=1}^{m}\left|Y_{j}^{\left(n\right)}-\langle Y\rangle \right|,$$
where 〈X〉 is the overall series mean. Then the absolute value scaling parameter $H_{AV}$ can be evaluated from the scaling relation
(68)$$\delta_{n}∼n^{H_{AV}-1}.$$

One more almost equivalent estimator of scaling properties regarding the FLSM is Higuchi’s method [11,175]. It relies on finding fractional dimension *D* of the length of the path. The normalized path length $L_{n}$ in this method is defined as follows
(69)$$L_{n}=\frac{N-1}{n^{3}}\sum_{i=1}^{n}\frac{1}{m-1}\sum_{j=1}^{m-1}\left|X_{i+jn}-X_{i+(j-1)n}\right|,$$
and $L_{n}∼n^{-D}$, where D=2−H.

We investigate four methods: AVE, Higuchi’s, MSD, and BDA for the analysis of ARFIMA time series as a test sample of FLSM.

### 4.2. Numerical Exploration of the Accumulated ARFIMA(0,d,0) Time Series

Let us consider the discrete process $\left\{X_{i}\right\}$ defined as a cumulative sum,
(70)$$X_{i+1}=X_{i}+Y_{i},$$
of correlated increments $\left\{Y_{i}\right\}$. Let the increments be generated by the ARFIMA(0,d,0) process [158,176]:(71)$$Y_{i}=\sum_{j=0}^{\infty }\frac{\Gamma (j+d)}{\Gamma (d)\Gamma (j+1)}Z_{i-j},$$
with random $Z_{i-j}$ from the domain of attraction of an α-stable law with 0<α≤2. One can calculate the sum in Equation (Equation 71) using the fast Fourier transform algorithm. The approximate relation between FLSM and ARFIMA can be derived using Riemann-sum approximation, see [176] for details.

Seeking to generate comparable time series with that analyzed in [165], the order disbalance time series of the financial markets we choose is $N=7\times 10^{6}$, nine values of d={−0.4,−0.3,−0.2,−0.1,0.0,0.1,0.2,0.3,0.4} and four values of α={2,1.5,1.25,1.0}. The sample time series for any set of parameters have been evaluated using four estimators described above: MSD, AVE, Higuchi’s estimator, and BDA. We evaluate *H* as described in the previous subsection. First of all, we partition time series $Y_{i}$ in subsets with $5\times 10^{5}$ time steps and accumulate them to obtain 14 subseries $X_{i}$. Then, the exponent λ or the Hurst parameter are evaluated for each subseries using MSD, AVE, and Higuchi’s sample estimators. Finally, we calculate the mean and standard deviation of defined 14 λ and *H* sets. Estimated *d* we calculate using d=H−1/α or d=(λ−1)/2 in MSD case. The graphs in Figure 8 of estimated *d* versus used ARFIMA model d serve as a good test of used estimators.

Our numerical result given in subfigure (**a**) confirms the theoretical prediction for the sample MSD $M_{N}\left(k\right)∼k^{2d+1}$ [156] as estimated *d* using this relation almost coincides with model *d* for all values of α. It is accepted that two estimators, absolute value and Higuchi’s, are almost equivalent and should be applicable for the analysis of fractional processes with stable distribution [11,167,168,174]. Indeed, the results of our numerical investigation, see (**b**) and (**c**) subfigures in Figure 8b,c, confirm the equivalence of these estimators. Nevertheless, the estimated values of memory parameter *d* deviate considerably from its model value, when α→1, and these deviations are much more prominent for the super-diffusion case d>0. These deviations do not arise as a computational effect, as the estimated relative standard deviation decreases from 0.15 to 0.02 for the evaluated *H* in the investigated interval of *d*. Fortunately, this result does not contradict the study [165], where we used these estimators to evaluate *d* in empirical order disbalance time series exhibiting sub-diffusion.

It is important to note that the estimators, MSD, AVE, and Higuchi’s should work well only for the unbounded time series when the most physical systems and processes are of finite size and duration. In all such cases, boundary effects might become important, and one must choose or propose more reliable estimators [167]. The BDA considered in our previous work [149,152,153,163], probably, can serve as an alternative approach. This method works better for the bounded time series, where more intersections of series with the threshold can be expected. Thus, in this contribution for the BDA, we restrict the diffusion of $X_{i}$ to the interval $\left[-X_{\max},X_{\max}\right]$ (in our analysis we use $X_{\max}={\left(10^{5}\right)}^{2d+1}$). This restriction is implemented as a soft boundary condition:(72)$$X_{i+1}=\max\left(\min\left(X_{i}+Y_{i},X_{\max}\right),-X_{\max}\right).$$

This iterative relation replaces Equation (Equation 70) in the $\left\{X_{i}\right\}$ series generation algorithm. We define the PDF of the burst and interburst duration $T_{j}$ for the whole set of time steps $N=7\times 10^{6}$ and the series threshold equal to zero mean. Note that only in this symmetric case PDF’s of burst and interbust duration coincide. Seeking to understand how the diffusion restriction mechanism impacts the results of other estimators, we use the same restriction mechanism for the 14 subseries obtained after the partition procedure. We present the results of this analysis in Figure 9.

Though the used diffusion restriction is relatively soft and changes the direction of movement in the limited number of trajectories points, the results of MSD, AVE, and Higuchi’s estimators changed very considerably—compare subfigures (**a**–**c**) with the corresponding results in Figure 9. Contrary, the results obtained using *H* defined by BDA, see subfigure (**d**), resembles AVE (**b**) and Higuchi’s estimator (**c**) subfigures from unbounded series Figure 9. Further investigation is needed to define the best methods and sample estimators for evaluating parameters of fractional time series impacted by various diffusion restrictions. The vast amount of data available from the financial markets can serve as empirical time series considered from the perspective of FLSM.

## 5. Future Considerations

Here, we have reviewed our approaches to modeling the long-range memory phenomenon and power–law statistics in a variety of complex systems. Our approach differs from the usual approach taken by mathematicians in that we have used Markovian models instead of the non-Markovian alternatives. We were able to reproduce similar behaviors due to our models being driven by various non-linear dependencies. In the case of SDEs, non-linearity may cause the increments of the stochastic process to be non-stationary and, by consequence, cause spurious long-range memory [177,178]. The many models we have built over the years are not models of true long-range memory; however, the critical question is whether our models capture the memory as observed in the financial markets and possibly other socioeconomic complex systems. Section 4, which describes our most recent endeavor, hints at three components that are needed to provide an answer.

The first component is a statistical test, which should distinguish between spurious and true long-range memory. Currently, we are considering the BDA method [148,149,150,151], which performs reasonably well in comparison to the alternatives. The core idea of the method is that for any one-dimensional Markovian random walk first-passage time PDF should be a power–law with exponent −3/2 at least for some of the duration. Deviations from this law could indicate the presence of true long-range memory. Though the method may fail when the stochastic process is not one-dimensional, the study of what happens in the multidimensional case, e.g., as in [99], is pending. Other challenges may also arise, as discussed in Section 4.

The second component would be a selection of models exhibiting both spurious and true long-range memory. Our prior research has introduced a variety of models of spurious long-range memory; hence, the next steps would be formulating comparable alternative models and studying properties of the existing long-range memory models. Here, we have focused on estimating long-range memory in the fractional Lévy stable motion (modeled using ARFIMA(0,d,0) discrete process), which is a generalization of the fractional Brownian motion; however, in general, other models could also be considered, for example, the multiplicative point process (see Section 2) could be generalized by replacing uncorrelated Gaussian noise with fractional Gaussian noise. Other correlation structures or variable pulse duration could also be considered as an extension [179]. Other notable alternatives and extensions include continuous-time random walk [180] and complex contagion frameworks [181,182].

The third component would be a variety of data from socioeconomic complex systems. Many of our earlier approaches relied on high-frequency absolute return and trading activity time series, but in our most recent works, we have shifted our attention to the order book data obtained from LOBSTER [164]. Order book data seem to invite a more general approach by understanding the data within FLSM or ARFIMA mindset for a broad class of anomalous diffusion processes [157,167,168]. The vast data in social and financial systems have to be investigated to identify and validate the fractional dynamics and long-range memory. Our first results in this direction [163,165] question the interpretation of long-range memory in the order flow data of financial markets. First of all, a prudent choice of estimators based on FLSM and ARFIMA assumptions are needed. After extensive analysis from this perspective, it would be possible to decide whether the investigated social system exhibits true long-range memory or observed power–law statistical properties are just the outcome of strong non-linear effects.

Research effort combining all these three components could yield a better understanding of the long-range memory phenomenon as it is observed in the variety of complex systems. The comprehensive interpretation of long-range memory observed in the financial and other social systems should considerably contribute to developing advanced analytical tools for applications in financial markets. Thus, we have focused on the description and explanation of the long-range memory phenomenon. Notably, a few more recent works refer to or use some of our results and are more application-minded. In [73] a non-linear SDE was derived, providing both physical and economic arguments, to study the performance of EUR/CHF exchange rate. The derived SDE belongs to the class described by (Equation 8). The author of [183] has considered the relationship between aging and long-range memory phenomena in a couple of physics experiments: blinking-quantum-dots, single-file diffusion, and Brownian motion in a logarithmic potential. The author of [184] has shown that SDE (Equation 8) applies to the modeling of the dynamics on microblogging networks. The author of [185] has considered the effects of perturbations on the stability of power–law distributions in general with an application to wealth distributions. The author of [186] tested the applicability of simple stochastic models to the modeling of non-stationary behavior of intraday tick-by-tick returns. The author of [187] has tested forecast robustness of non-linear GARCH model when time series exhibit high positive autocorrelation. Mean reversion phenomenon was studied in Karachi Stock Exchange data from the perspective of GARCH models in [188]. The authro of [189] has compared the performance of non-linear SDE models against Black and Scholes model, which is one of the models used by the practitioners. Various modifications of Heston model, another model favored by the practitioners, are also reminiscent of SDE (Equation 8) [190]. We hope to inspire and maybe take up more application–minded endeavors.

## Figures and Tables

**Figure 1 entropy-23-01125-f001:**
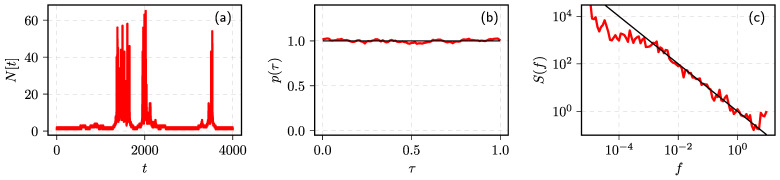
Statistical properties of the point process by numerically solving Equation (Equation 2): (**a**) sample fragment of corresponding $N\left[t\right]$ time series, (**b**) PDF of the inter-event times, and (**c**) PSD of the process. Red curves correspond to numerical results, while black curves are theoretical power–law fits with (**b**) α=0 and (**c**) β=1. Model parameter values: γ=0, μ=0, σ=0.1, w=1.

**Figure 2 entropy-23-01125-f002:**
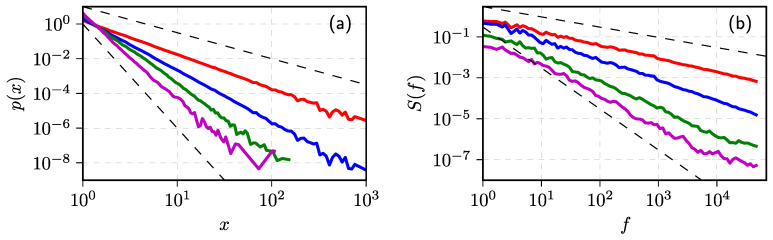
Various slopes of PDF (**a**) and PSD (**b**) reproduced by the numerical solutions of SDE (Equation 8). Model parameter values: σ=1, η=2.5 (all cases) and λ=2 (red curves in both (**a**,**b**)), 3 (blue curves), 4 (green curves), and 5 (magenta curves). Black dashed lines correspond to (**a**) $p\left(x\right)∼x^{-\lambda }$ with λ=1.5 and λ=6 (upper and lower curves), (**b**) $S\left(f\right)∼1/f^{\beta }$ with β=0.5 and β=2 (upper and lower curves).

**Figure 3 entropy-23-01125-f003:**
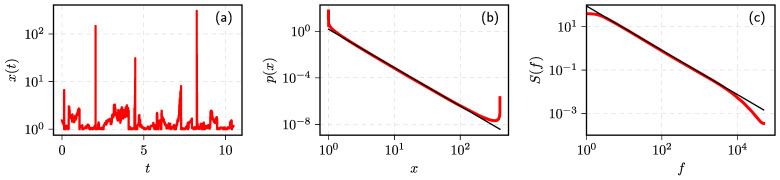
Statistical properties of the time series obtained by solving SDE with Lévy α-stable noise, Equation (Equation 16): (**a**) sample fragment of the time series, (**b**) PDF, and (**c**) PSD of time series. Red curves correspond to numerical results, while black curves are power–law best fits with exponents (**b**) λ≈3.3, (**c**) β≈1.

**Figure 4 entropy-23-01125-f004:**
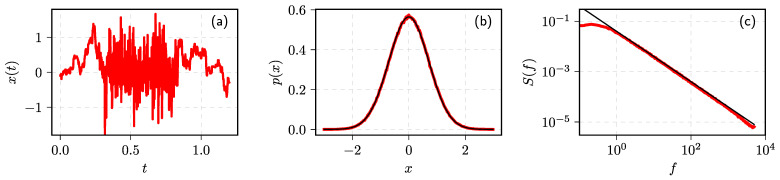
Statistical properties of the time series obtained by solving coupled SDEs (Equation 39) and (40): (**a**) sample fragment of x(t) time series, (**b**) PDF of the externally observed values *x*, and (**c**) PSD of x(t). Red curves correspond to numerical results, while black curves are theoretical fits: (**b**) standard Gaussian PDF, (**c**) $S\left(f\right)∼1/f^{\beta }$.

**Figure 5 entropy-23-01125-f005:**
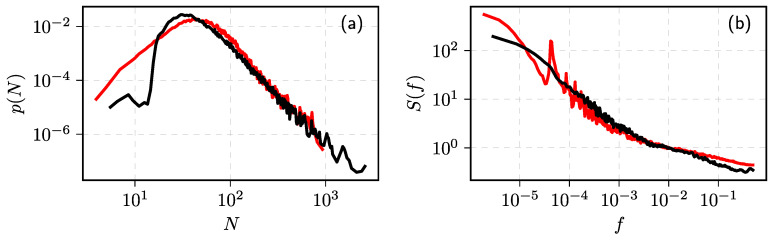
Trading activity (**a**) PDF and (**b**) PSD for MMM stock traded on NYSE (red curve) and the numerical solutions of SDE (Equation 42). Model parameters values: η=2.5, λ=4.3, $\sigma^{2}=0.045$, ϵ=0.36, $n_{0}=0.14$. Empirical and numerical PDF was obtained by considering trades in the 300s time window.

**Figure 6 entropy-23-01125-f006:**
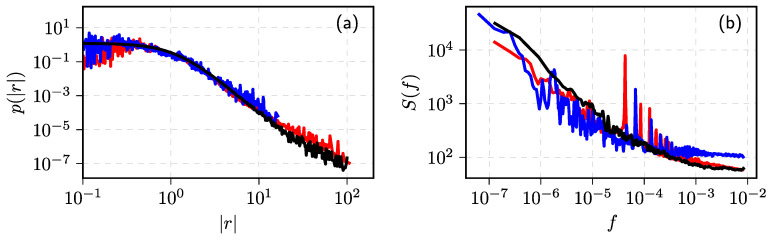
Comparison of empirical (**a**) PDFs and (**b**) PSDs of absolute one minute return as observed in NYSE (red curves) and VSE (blue curves) stocks. Empirical results are compared against the model, generated by the SDE (Equation 43) and exogenous noise Equation (Equation 44), (black curves). Model parameter values: η=2.5, λ=3.6, ϵ=0.017, $x_{\max}=10^{3}$, $\lambda_{2}=5$.

**Figure 7 entropy-23-01125-f007:**
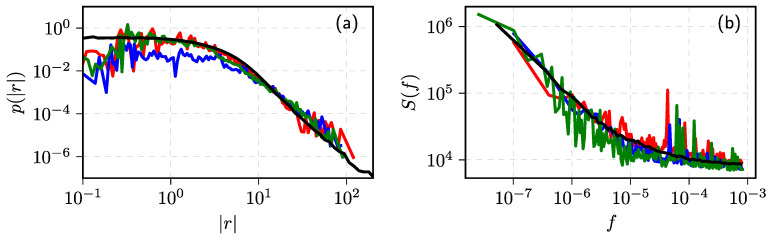
Comparison of empirical (**a**) PDFs and (**b**) PSDs of absolute ten minute return as observed in NYSE (red curves), VSE (blue curves), and WSE (green curves) stocks. Empirical results are compared against the consentaneous model, defined in [41]. Model parameter values are the same as in Figure 2 of [41].

**Figure 8 entropy-23-01125-f008:**
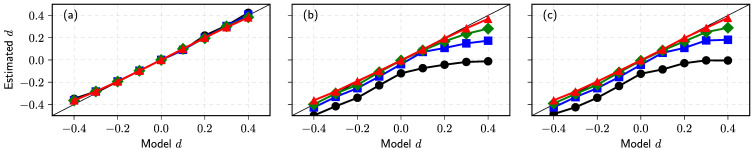
Comparison of the MSD (**a**), AVE (**b**), and Higuchi (**c**) estimator performance when estimating *d* from the accumulated ARFIMA(0,d,0) series in the unbounded case, $\left\{X_{i}\right\}$ generated by Equation (Equation 70). Different curves correspond to the different values of the noise distribution stability parameter: α=2 (red triangles), 1.5 (green diamonds), 1.25 (blue squares), and 1 (black circles).

**Figure 9 entropy-23-01125-f009:**
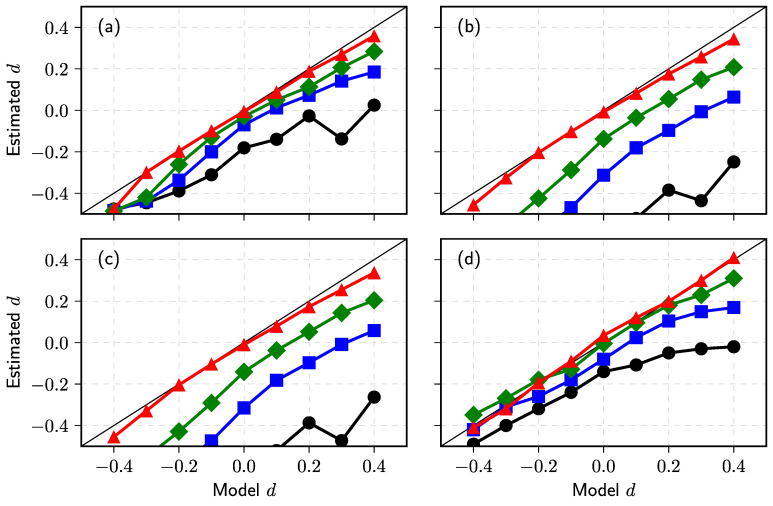
Comparison of the MSD (**a**), AVE (**b**), Higuchi (**c**), and BDA (**d**) estimator performance when estimating *d* from the accumulated ARFIMA(0,d,0) series in the bounded case, $\left\{X_{i}\right\}$ generated by Equation (Equation 72). Different curves correspond to the different values of the noise distribution stability parameter: α=2 (red triangles), 1.5 (green diamonds), 1.25 (blue squares), and 1 (black circles).

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
