# Peer review of "Understanding the Nature of the Long-Range Memory Phenomenon in Socioeconomic Systems"

_entropy, 2021, doi:10.3390/e23091125_

Round 1

Reviewer 1 Report

This paper is interesting. The observation I make is that some literature related to econophysics is missing, so the following suggestions are followed:

- Pereira, E. J. D. A. L., da Silva, M. F., & Pereira, H. D. B. (2017). Econophysics: Past and present. Physica A: Statistical Mechanics and its Applications473, 251-261.

- Jovanovic, F., & Schinckus, C. (2017). Econophysics and financial economics: An emerging dialogue. Oxford University Press.

Author Response

Thank you, the suggestion to include additional citations related to econophysics is accepted, see the revised Introduction.

Reviewer 2 Report

The topic of the study is interesting: long-range memory in  a class of complex systems connected to the society. The idea of study is also interesting: to reproduce the long-range memory by Markov processes. The study is directed by the idea of application of results to complex systems connected to financial markets.  The introduction section is devoted to the history of the research of the authors, and section 2 contains basics and results about the modeling by multiplicative point processes, stochastic differential equations, presence of anomalous diffusion in the long-range memory models, and other theoretical tools for modeling the statistical properties of the financial markets.  Another line in the overview is connected to the agent-based models of the long-range memory in the financial markets. Several models are discussed which are probably of interest for the research work of the authors. Application of these models to the problems from the area of opinion dynamics is also presented. This is an interesting addition tho the main discussion connected to the financial markets. Finally, the authors describe several ideas about the search for appropriate long range memory test. The question is important as the efforts of authors to model the long-range memory effects are based on Markov processes which is different from other much used approaches bases on non-Markov processes.

In summary and especially because of the last remark above, I think that the manuscript will be  intersting for a large group of researchers who have interest in stochastic processes, agent-based models and their application to economic and social systems. Because of this my opinion about the publication of the manuscript is positive.

Author Response

Thank you for the very positive comments.  

Reviewer 3 Report

The authors summed up their research on the financial time series using different methods and models. In the introduction, they motivated their paper by some memories of how their works started and continued. The paper is a survey of a vast extent of methodology. Some illustrative examples are presented as well.  It is interesting and worth reading.

However, from the perspective of almost thirty years of developing the econophysics, the reader can expect a more efficient survey  from the perspective of long memory models developed in empirical finance.

The aim is as follows: “Here we provide an overview of our approach to understanding and modeling the 67 long-range memory phenomenon in financial markets and other complex systems 68 and share our most recent result.” I would suggest convincing the reader that apart from a vast range, there is a similar value-added in these approaches with specific references to how much the proposed methods were applied in practice or in what way they improved analytical tools for financial markets. A comparison with other methods or approaches would be welcome.

Author Response

Thank you for the valuable comments and suggestions. Our contribution to econophysics is probably not so extensive as many practitioners of finance might expect. Nevertheless, the comprehensive understanding of the nature of the long-range memory phenomenon in socio-economic systems seems mandatory in developing related analytical tools for financial markets. We hope that our work and results will be helpful for the further development of econophysics and practical tools. Following your recommendation, in the revised version of our manuscript, we included citations to much broader reviews on the subject in the Introduction and added the citations to our work at the end of the concluding section.